# Irradiation-Induced Amorphous-to-Crystalline Phase Transformations in Ceramic Materials

**DOI:** 10.3390/ma15175924

**Published:** 2022-08-27

**Authors:** Cyrus Koroni, Tristan Olsen, Janelle P. Wharry, Hui Xiong

**Affiliations:** 1Micron School of Materials Science & Engineering, Boise State University, Boise, ID 83725, USA; 2School of Materials Engineering, Purdue University, West Lafayette, IN 47906, USA; 3Center for Advanced Energy Studies, Idaho Falls, ID 83401, USA

**Keywords:** amorphous-to-crystalline transformations, ceramics, nuclear stopping, electronic stopping, irradiation damage, microstructural evolution

## Abstract

Amorphous ceramics are a unique class of materials with unusual properties and functionalities. While these materials are known to crystallize when subjected to thermal annealing, they have sometimes been observed to crystallize athermally when exposed to extreme irradiation environments. Because irradiation is almost universally understood to introduce disorder into materials, these observations of irradiation-induced ordering or crystallization are unusual and may partially explain the limited research into this phenomenon. However, the archival literature presents a growing body of evidence of these irradiation-induced amorphous-to-crystalline (a-to-c) phase transformations in ceramics. In this perspective, the summary and review of examples from the literature of irradiation-induced a-to-c transformations for various classifications of ceramics are provided. This work will highlight irradiation conditions and material parameters that appear most influential for activating a-to-c transformations, identify trends, examine possible mechanisms, and discuss the impact of a-to-c transformations on material properties. Finally, future research directions that will enable researchers to harness a-to-c transformations to tailor materials behaviors will be provided.

## 1. Introduction

Ceramics are exposed to extreme irradiation environments in numerous applications, including nuclear fuels and claddings, immobilizing and storing nuclear waste [1,2,3], radiation shielding, space exploration and travel, and doping of semiconductors [4,5,6,7,8,9,10,11,12,13]. In such environments, energetic particles impinging on crystalline ceramic materials lead to the creation of damage through various energy transfer mechanisms, culminating in changes to material microstructures, properties, and performance [14]. Although crystalline ceramics have historically been used in practical irradiation-facing applications, amorphous ceramics have recently garnered interest for their unique properties and characteristics, including irradiation tolerance.

Amorphous materials are purported to better resist irradiation than crystalline ceramics. But while irradiation effects in crystalline ceramics have been widely studied and are thus moderately understood, radiation effects in amorphous materials have received far less attention, and the underlying mechanisms remain largely unknown. Because the irradiated atomic arrangements and microstructures can influence the originally designed material properties, morphology, and functionality, understanding these underlying material responses to irradiation is critical.

In a crystalline ceramic, the collisions between target atoms and irradiating particles displace target atoms from their lattice positions and consequently generate point defects such as vacancies and self-interstitials. These point defects can remain in the material matrix, agglomerate into vacancy or interstitial clusters, or annihilate with one another at defect sinks such as grain boundaries, interfaces, or free surfaces. In crystalline ceramics, the accumulation of point defects in the matrix often results in the disordering and amorphization of the target material, and this disordering can be localized or global [15,16,17,18,19,20,21,22,23,24,25]. The resultant irradiated microstructure is often a composite product containing accumulated point defects, amorphous regions, point defect clusters, extended defects such as dislocation loops or voids which form due to defect cluster growth, chemical segregation and precipitation, and phase transformations. Microstructural changes depend on experimental parameters such as the irradiating specie, the incident particle energy, irradiation temperature, the irradiation flux (i.e., dose rate), and the total fluence (i.e., dose) [26].

The natural lack of order or structure in amorphous ceramics inherently provides the basis of their apparent irradiation tolerance since it is extremely difficult to distinguish an irradiation-induced “point defect” within pre-existing disorder. However, instead, irradiation-induced crystallization, phase nucleation, and ordering have recently been reported in amorphous ceramic materials. While amorphous-to-crystalline (a-to-c) transformations tend to occur in ceramic materials under thermal annealing, the literature reveals that irradiation can induce a-to-c transformations athermally. However, the underlying mechanisms behind irradiation-induced athermal a-to-c transformations are not well understood. In this perspective, the summary and review of examples in the literature will be included for different categories of ceramics where a-to-c transformations have been observed, highlighting irradiation parameters and conditions that appear to be most conducive to creating the a-to-c transformation, and discussing possible mechanisms. This perspective excludes semiconductor materials for electronic devices, as their irradiation-induced transformations have been extensively summarized in other accounts and studies [27,28,29,30].

## 2. Energy Dissipation Mechanisms

The ionic nature of ceramic materials underscores the importance of understanding how incident irradiating particles interact with both target nuclei and their electrons. Thus, the basic energy dissipation mechanisms that lead to the creation of individual point defects during irradiation are reviewed here. Irradiation of solid matter involves the transfer of kinetic energy from the incident particles (which can have kinetic energies ranging from ~eV to ~GeV depending on the application) to the target solid. The irradiating particles lose their kinetic energy through collisions and interactions with atoms in the target material system [31,32]. The energy loss of energetic particles in matter is commonly described as stopping power (S(E)), which is described by [32]:(1)$$S\left(E\right)=-\frac{\delta E}{\delta x}=-{\left(\frac{\delta E}{\delta x}\right)}_{nucl.}-{\left(\frac{\delta E}{\delta x}\right)}_{elec.}-{\left(\frac{\delta E}{\delta x}\right)}_{rad.}$$
where $\frac{\delta E}{\delta x}$ is the differential energy loss per unit length of travel of the irradiating particle and is comprised of nuclear, electronic, and radiative contributions [32]. The radiative term is negligible in the types of studies examined in this perspective, rendering energy transfer mechanisms in ceramics mainly influenced by nuclear and electronic stopping.

The energy transfer may be categorized into three different regimes [33], as shown in Figure 1. At lower projectile energies, the transfer of energy to atomic nuclei tends to dominate the interactions, leading to atomic displacements within the solid. At higher projectile energies, interactions with the electrons in the solid will be the dominant mode of energy dissipation, often creating ion tracks within the target. Finally, at intermediate energies, both nuclear and electronic energy losses occur [34,35,36], causing additive effects on damage production or competitive recovery processes that affect damage accumulation and evolution.

During nuclear stopping, elastic collisions between the incident ion and target atoms result in atomic displacements, which lead to point defect formation. This process can be described by the binary collision approximation (BCA) model [37], in which a sequence of independent collisions occurs between the incident and target species, and the energy transfer is dependent upon the interatomic potential. The collective sequence of collisions is known as a collision cascade. Cascades introduce point defects (vacancies and interstitials), and their diffusion ultimately controls the structural, microstructural, and phase evolution of the material.

With increasing incident ion energy, electronic stopping, i.e., inelastic collisions between the bound electrons in the target material and the projectile ion (including projectile electrons), becomes a more dominant mode of energy transfer. Electronic energy loss mechanisms can be described by the thermal spike model [38,39]. In the thermal spike model, electrons along the incident ion path undergo excitation, resulting in energy transfer to the surrounding atoms through electron-phonon coupling and causing localized heating [33]. This phenomenon can produce residual ion tracks in the target material, particularly when significant localized temperature changes and energy transfer occur. During de-excitation, recoil energies can be sufficiently high so as to cause displacements of target atoms.

## 3. Review of Observed A-to-C Transformations

Materials wherein a-to-c transformations have been observed in a variety of compositions and under a variety of irradiating conditions are summarized in Table 1. Important parameters that likely affect the observed a-to-c transformation, such as the target material, irradiating particle species, the energy of the irradiating beam, fluence, and irradiating temperature, are listed in Table 1 where relevant. The a-to-c transformations can be broadly categorized into three types: (1) athermal, resulting exclusively from either electron or ion irradiation of amorphous material, (2) resulting from the combined effects of heating and either ion or electron irradiation, or (3) epitaxial growth or recrystallization of an amorphous oxide due to ion or electron irradiation. The a-to-c transformations are usually observed in target materials that are amorphous as-prepared [40,41,42,43,44,45,46,47]. Less commonly, crystalline or single crystal materials are first amorphized by one type of irradiation, and then the a-to-c transformation is accomplished by irradiating the material in a subsequent experiment [48,49,50,51]. This section summarizes reported a-to-c transformations in ceramic materials, including zirconium oxide, titanium oxide, aluminum oxide, silicon oxide materials, silicon carbide, silicates, and other ceramic materials.

### 3.1. Zirconium Dioxide

Roddatis et al. demonstrated that transmission electron microscopy (TEM) beam irradiation of amorphous ZrO_2_ films prepared via a self-assembled monolayer method can result in an a-to-c transformation [40]. At a cryogenic temperature of 4.2 K, a 160 kV TEM electron beam at a current density of 0.5 A/cm^2^ was able to form crystallites in the as-prepared amorphous film. The size and number of crystallites increased with irradiation exposure time. After 20 min of TEM electron beam irradiation, the crystallites grew to ~10 nm in size, and fast Fourier transform (FFT) analysis of the high-resolution TEM (HR-TEM) images confirmed that the crystallites were structured as tetragonal ZrO_2_. To assess whether there was a change in the coordination and film composition as a result of TEM electron irradiation, electron energy loss spectroscopy (EELS) was performed on an as-prepared amorphous sample, a sample irradiated for 5 min with TEM electrons, a tetragonal ZrO_2_ powder, and a monoclinic ZrO_2_ powder. From the oxygen K-edge spectra, it was found that the amorphous sample lacked a distinct peak at 534 eV, but such a peak was present in both the irradiated sample and the reference standards. Therefore, the EELS results provided evidence for a change in coordination in the amorphous film due to electron irradiation-induced a-to-c transformation. It was also found that the a-to-c transformation occurred faster when amorphous films were irradiated with 200 kV electrons at room temperature compared to with 160 kV electrons at 4.2 K. However, crystallization at room temperature was faster with 200 kV than with 400 kV electron irradiation. While the effects of temperature and electron energy could not be deconvoluted, the authors explained that this seemingly strange result might be due to inelastic scattering being less probable at higher electron energies and that inelastic electron scattering of electrons on the sample surface plays a role in the electron irradiation-induced structural changes observed [40].

Naguib and Kelly irradiated amorphous ZrO_2_ films with Kr ions at energies ranging from 2–35 keV [48]. The amorphous films were prepared by sputtering crystalline ZrO_2_ films on Zr foil with Kr or O_2_ ions at 6 keV for 75 min, yielding an amorphized layer of ZrO_2_. The as-prepared films were confirmed to be amorphous by TEM selected area electron diffraction (SAED). They first investigated the thermally-induced crystallization of amorphous ZrO_2_ by TEM in situ pulse heating over 400–540 °C. This was found to result in recrystallization of the amorphous film, and SAED analysis indicated that the resulting crystallites were mainly cubic, with some monoclinic character (inset image in Figure 2a) [48]. As a control, amorphous films were heated in a tube furnace until crystallization was achieved. Samples achieved full crystallization when heated at temperatures between 520–540 °C for 5 min or in air at 400–425 °C for 360 min and showed similar SAED patterns to the pulse-heated sample.

Subsequently, Naguib and Kelly observed athermal irradiation-induced a-to-c transformations as a result of Kr ion bombardment at fluxes ranging from 1–20 µA/cm^2^. As the dose rate (i.e., flux) decreased, the corresponding threshold ion fluence required to induce an a-to-c transformation increased dramatically. The crystallites produced by ion bombardment were found to produce SAED patterns similar to those produced in crystalline films formed by heating alone and yielded crystallite sizes of 80–100 nm and ~700 nm, respectively (Figure 2b).

To determine whether the observed crystallization was primarily influenced by local heating from the ion beam, a 400-mesh Cu grid was placed loosely on the surface of an amorphous sample prior to irradiation (Figure 2b). At low flux (<7 µA/cm^2^), TEM and SAED analysis showed in the top right inset of Figure 2b that the regions shielded from ion bombardment (region ‘A’) remained amorphous while the unshielded regions underwent the expected a-to-c transformation. This is evidenced by the shielded region (Figure 2b, region marked ‘A’) producing a SAED pattern with diffuse and indistinct rings, while the unshielded region (Figure 2b, region marked ‘B’) exhibited a distinct diffraction pattern that could be attributed to cubic ZrO_2_. It was also found that irradiation at higher fluxes (20 µA/cm^2^) resulted in crystallization of both the shielded and unshielded regions. Crystallites in the unshielded region ‘B’ were larger than those in the shielded region and resembled the crystallites induced by pulse heating (e.g., those in Figure 2a) [48]. This implied that at low flux, the a-to-c transformation is primarily athermal and caused by ion bombardment alone, whereas, at higher flux, the influence of local heating may be more significant.

In a similar study, Leteurtre and Soullard created amorphous ZrO_2_ films prepared by sputtering crystalline ZrO_2_ with O_2_ ions [49]. The resulting amorphous films were then irradiated with 1 MeV Kr ions. While these irradiation conditions did not consistently result in a homogeneous a-to-c transformation, TEM analysis of the irradiated amorphous films exhibited distinct diffraction spot patterns in regions having higher electron transmission, indicating at least partial a-to-c transformation. The authors attributed this to the effects of collision cascades creating zones with differentiated order.

In a more recent study, Lian et al. reported ion irradiation-induced a-to-c transformation in thin bilayer films of nanocrystalline cubic and amorphous ZrO_2_ layers [41]. Room temperature TEM in situ ion irradiation was performed using 1 MeV Kr^+^ ions at a flux of 6.25 × 10^10^ ions/cm^2^·s to various total fluence levels. TEM and SAED analysis of the bilayer provided evidence of an amorphous-to-tetragonal phase transformation in the original amorphous layer (Figure 3). Specifically, evidence of recrystallization was observed from TEM analysis after irradiating the film to a fluence of 3.13 × 10^14^ ions/cm^2^ and a *d_102_* ring corresponding to tetragonal ZrO_2_ was observed after irradiating to a fluence of 3.13 × 10^14^ ions/cm^2^, 1.88 × 10^15^ ions/cm^2^, and 3.13 × 10^15^ ions/cm^2^ [41]. Because this a-to-c transformation occurred at room temperature and no significant ion beam heating of the sample occurred under the irradiation conditions used, the authors suggested that the a-to-c transformation might be more energetically favorable under non-equilibrium conditions than under thermal conditions alone [41]. They also reported that irradiation-induced grain growth was more prominent for tetragonal crystallites that formed in the amorphous layer than for cubic grains in the nanocrystalline layer, but that the grain growth in both layers was strongly dependent on irradiation fluence [41].

### 3.2. Titanium Dioxide

Kern et al. conducted experiments in which electrolytically deposited amorphous TiO_2_ films on polished AISI 316 steel substrate were irradiated with a scanning electron microscope (SEM) electron beam at 20 kV under beam currents of 24 µA, 250 nA, 30 nA, and 10 nA [42]. The samples were mounted to a heating stage that was either held at room temperature or 250 °C, and the irradiation time was varied between 1 and 600 s. Results showed evidence of a decrease in film thickness by ~20 nm when irradiated at 10 nA and by ~40 nm when irradiated at 30 nA or higher beam current. The change in film thickness was the result of electron-stimulated O desorption in the amorphous TiO_2_ film and reduction of TiO_2_ to TiO [42]. Some of the irradiation conditions resulted in a phase transformation from an amorphous state to anatase TiO_2_ as identified via Raman spectroscopy (Figure 4). Specifically, the amorphous-to-anatase transformation occurred locally after irradiation with 24 µA for 600 s at room temperature. Increasing temperature accelerated the rate of transformation; when the sample stage was preheated to 250 °C, the transformation occurred after only 60 s at the same beam current. Both experimental temperatures were below the 440 °C thermally-induced crystallization temperature of electrolytically deposited amorphous TiO_2_ films [42,62]. Local heating from the electron beam may have contributed to, but could not completely explain, the lower phase transformation temperatures under irradiation.

Smith and coworkers investigated the effects of proton irradiation on amorphous TiO_2_ nanotubes and found that 200 keV proton irradiation at a fluence of 2.18 × 10^17^ ions/cm^2^ at both room temperature (RT) and a higher temperature (HT) of 250 °C resulted in varying degrees of a-to-c phase transformations [43]. TEM and SAED analysis (Figure 5) showed that irradiation at RT resulted in a partial phase transformation of the amorphous material to anatase and rutile TiO_2_, while irradiation at 250 °C resulted in the formation of only the disordered rutile phase. Rings corresponding to anatase (101) and rutile (101) and (111) were indexed in the RT irradiated sample (Figure 5f), while only rings corresponding to the rutile phase were visible in the SAED for the HT sample (Figure 5i). The presence of a disordered rutile phase was also confirmed by Raman spectroscopy and EELS. Based on the phase diagram of TiO_2_, the crystallization to the anatase phase generally requires thermal annealing temperatures > 400 °C, and the rutile phase requires temperatures > 600 °C [43,63]. In a sense, this corroborates Kern’s [42] finding that increasing irradiation temperature accelerated the a-to-c transformation. Additionally, thermal annealing of TiO_2_ nanotubes at >600 °C is generally not a viable method for synthesizing rutile TiO_2_ nanotubes because it results in the collapse of the nanotube structure [43]. However, SEM analysis showed that the proton irradiation did not significantly change the morphology of the nanotubes (Figure 6). Therefore, this work highlights a new avenue to create crystalline nanostructures through ion irradiation on amorphous starting materials.

In another study, Yang and coworkers [44] investigated the effects of TEM in situ 46 keV Au^–^ ion irradiation on amorphous and anatase TiO_2_ nanotubes at ambient temperature to a fluence of 2.3 × 10^14^ ions/cm^2^. In some amorphous TiO_2_ nanotubes, it was observed that bending of the nanotubes occurred as a result of irradiation of Au^–^ ions under these conditions, while anatase nanotubes exhibited no significant change in morphology (Figure 7). While some changes in contrast were observed on the anatase nanotubes during irradiation (Figure 7b, areas circled in red), the features were too small to identify as defects or phase transformations.

This morphological change of the initially amorphous nanotubes was attributed to partial irradiation-induced a-to-c transformation, as observed via HRTEM. Sites of partial a-to-c transformation in the amorphous TiO_2_ nanotubes tended to be more prevalent near areas where the nanotubes were bent. The authors proposed that the heterogeneous formation of nanocrystal domains rather than complete crystallization could be best explained by inspection of energy loss mechanisms and ion range profiles for TiO_2_ (Figure 8).

The extent of irradiation-induced crystallization, and therefore the size of resulting crystallites, decreased with increasing electronic stopping power. Additionally, the magnitude of either nuclear or electronic stopping power varied depending on the orientation of nanotubes relative to the incident ion beam since the nanotubes were randomly distributed on the TEM grid (Figure 8a). Furthermore, stopping and range of ions in matter (SRIM) simulations predicted that peak Au ion implantation occurred at a depth of about 30 nm, and since the thickness of the nanotube walls was about 10 nm, many incident ions would pass through the nanotubes if they were oriented normal to the beam direction (Figure 8b) [44].

### 3.3. Aluminum Oxide

Liu et al. investigated the effects of electron irradiation of amorphous Al_2_O_3_ that led to an a-to-c transformation [45,46]. The as-prepared samples were 300 nm thick layers of amorphous Al_2_O_3_ on dislocation-free 〈100〉 B-doped Si wafers. Electron irradiation was conducted using a TEM electron beam operating at 100 keV at a dose rate of 100 mA/cm^2^. When the exposure time was varied over 1–16 min at a dose rate of 100 mA/cm^2^, near-spherical trigonal Al_2_O_3_ crystallites formed in the bulk layer of Al_2_O_3_, and dendrites formed at the Al_2_O_3_/Si boundary. As irradiation time increased, the size and number of crystallites in the amorphous layer increased, while the dendrites remained unchanged. The size of crystallites began to approach a steady state after about 5 min, and the crystallite density plateaued after an irradiation time of 10 min. A critical electron beam dose rate was required for the a-to-c transformation to occur. Specifically, when the particles were irradiated at 10 mA/cm^2^, no a-to-c transformation occurred [45]. The authors initially thought that thermal activation from the electron beam due to local heating was the main cause of the a-to-c transformation [45]. However, in a follow-up publication, the estimated temperature contribution was found to not be significant enough to be a factor in the mechanism for the observed a-to-c transformation [46]. The authors subsequently proposed that electron irradiation-induced defects were a more likely cause for the a-to-c transformation [46].

Zhou and Sood used two ion irradiation conditions to first form an amorphous layer in an α-axis oriented single crystal sapphire (α-Al_2_O_3_), then subsequently induce epitaxial crystallization in that amorphous layer [50]. Amorphous layers on single crystal α-Al_2_O_3_ samples were generated by irradiating with 100 keV In ions to a fluence between 0.7 and 2.7 × 10^16^ ions/cm^2^. The amorphized layers were then irradiated with 1.5 MeV Si ions to a fluence of 3 × 10^16^ ions/cm^2^ at 400 °C. Rutherford backscattering and channeling (RBSC) measurements showed that a thin amorphous layer was successfully created on the surface of the α-Al_2_O_3_ after the initial In ion irradiation. RBSC spectra after the 400 °C Si irradiation, then showed signs of recrystallization to a γ-Al_2_O_3_ phase, consistent with that formed through 700 °C thermal annealing. These results demonstrated that epitaxial recrystallization of amorphous Al_2_O_3_ is possible at lower temperatures when used in conjunction with Si ion irradiation.

In a more recent study of amorphous films on single crystal (0001) α-Al_2_O_3_, Ning et al. found that epitaxial growth of γ-Al_2_O_3_ in the amorphous layer could be induced via irradiation with either 360 keV Ar^2+^ ions or 180 keV O^+^ ions [47] at various temperatures. RBSC spectra of the amorphous films before and after irradiation indicated that epitaxial recrystallization of γ-Al_2_O_3_ had taken place, and this was confirmed by cross-sectional TEM and SAED. When comparing SAED patterns of the unirradiated and O ion irradiated films, a diffraction pattern appeared that was consistent with orientation-related epitaxial growth of γ-Al_2_O_3_ from the original single crystal layer [47]. The ion beam-induced epitaxial growth between 400 and 600 °C produced films with higher epitaxial quality than those produced by annealing alone at 800 °C. This improvement in quality was attributed to the elimination of {111} twin boundaries in the γ-Al_2_O_3_ layer due to ion channeling effects during the irradiation.

Finally, Sina et al. showed that epitaxial crystallization of γ-Al_2_O_3_ on α-Al_2_O_3_ could be accomplished via two-step ion irradiation at room temperature [51]. In their study, α-Al_2_O_3_ samples were irradiated with Zr^+^ ions at 175 keV to fluences ranging from 7.5 × 10^15^ to 1.5 × 10^16^ ions/cm^2^ and then subsequently irradiated with O^+^ ions at 55 keV to fluences that corresponded with 25% of the maximum displacement damage dose produced by the initial Zr^+^ ion irradiation. The lower Zr^+^ ion fluence (7.5 × 10^15^ ions/cm^2^) specimens had buried polycrystalline (but not amorphous) regions at depths that corresponded to the maximum Zr^+^ ion range calculated by SRIM; subsequent irradiation with O^+^ ions changed little. Meanwhile, in samples that were irradiated to higher Zr^+^ ion fluences of 1.5 × 10^16^ ions/cm^2^, a sub-surface amorphous layer appeared at depths of 20–65 nm, which corresponded to the peak damage calculated by SRIM simulations (Figure 9) [51].

Subsequent irradiation with O^+^ ions showed evidence of epitaxial recrystallization at the interface between the buried Zr^+^ irradiation-induced amorphous region and the crystalline surface region (Figure 9). Analysis was conducted of the crystalline near-surface region, the buried amorphous region, and the bottom-most crystalline region using nano-beam electron diffraction (NBED). The NBED analysis found that the bottom-most region was consistent with the diffraction pattern for α-Al_2_O_3_. The buried amorphous region exhibited diffuse rings, consistent with an amorphous phase. However, the near-surface region contained diffraction patterns consistent with both α-Al_2_O_3_ and γ-Al_2_O_3_, suggesting that epitaxial γ-Al_2_O_3_ can form through two-step irradiation-induced amorphization and subsequent recrystallization.

### 3.4. Silicon Oxide

Upon ion irradiation, a-to-c transformations have been observed in the Si regions of SiO after SiO transformed into both Si and SiO_2_ [52,53]. Rodichev et al. [52] irradiated SiO at room temperature with 575 MeV Ni and 863 MeV Pb ions to fluences of 10^11^ ions/cm^2^ and 10^13^ ions/cm^2^, respectively. They observed the transition of SiO to Si and SiO_2_ through infrared (IR) spectroscopy, in which the Si-O bond for SiO at 1000 cm^−1^ shifted to 1035–1055 cm^−1^, which was consistent with the Si-O bond for SiO_2_. TEM images of the irradiated samples confirmed these results with two distinct regions observed: one with light-contrasting amorphous zones and the other being localized dark-contrasting zones attributed to high-Z number contrast. Si nanocrystals were believed to be present due to the lattice fringes in the dark-contrast regions within the HR-TEM images. These darker regions were located in a track within the material in a crystal-like arrangement. The number of dark regions increases with ion fluence. The authors concluded that two types of transformations occurred. The first involved short-range diffusion where SiO converted to Si and SiO_2_, wherein the Si aggregates were small and mixed with SiO_2_ and were thus unresolvable in TEM. The second type of transformation that occurred was diffusion over several atomic distances close to the ion track, which accounted for the oxygen concentration near the core of the ion track and rearrangement of Si atoms into nanocrystals.

When amorphous SiO was irradiated with 80 keV He ions at a fluence of 7 × 10^20^ ions/cm^2^, small circular precipitates were observed by Walters et al. [53]. The precipitates exhibited an electron diffraction pattern matching closely with crystallized Si. During irradiation, the sample reached 850 °C. To ensure that the crystallization was an effect of irradiation and not thermal annealing, the researchers thermally annealed the samples at 900 °C. Following 60 min exposure to 900 °C, there were no changes, suggesting that the crystallites formed as a result of irradiation. Walters and coworkers also found that the crystallization as a function of dose was non-linear and instead followed an S-shaped curve described by the Avrami equation. Below a flux of 2 × 10^18^ ions/m^2^·s, the authors state that, similar to other non-metallic materials, amorphous SiO requires extended diffusion to form the observed precipitates.

SiO_2_ has been investigated with neutral atom, ion, and electron irradiation [12,13], and a-to-c transformations were found with neutral atom and electron irradiation. When Mizutani et al. irradiated SiO_2_ with 400 eV Ne neutral atoms or Ne ions to a fluence of 2 × 10^17^ atoms/cm^2^ or ions/cm^2^, RBS showed that O was preferentially sputtered by the Ne ion, creating an O-deficient, Si-rich layer on the surface of the ion irradiated sample [12]. By contrast, neutral Ne atom irradiation did not create a chemical deficiency or enrichment at the surface. Auger electron spectroscopy (AES) showed that the neutral Ne atom irradiated sample changed to a different phase, although its chemical composition was maintained, whereas the Ne ion irradiated sample showed AES peaks associated with Si and SiO_x_, indicating chemical separation.

When neutral Ne atom irradiation was conducted at a lower energy of 350 eV and to a lower fluence of 1 × 10^17^ atoms/cm^2^, TEM analysis exhibited darker regions, suggesting crystalline phases were formed, though electron diffraction patterns could not be obtained due to the small size of the crystallites. Using Ne ions at the same energy of 350 eV to the same fluence of 1 × 10^17^ ions/cm^2^, however, no layers or regions having different contrast were found, even with the Si-rich surface layer. Furthermore, following neutral Ne and Kr atom irradiation (both at 350 eV, 1 × 10^17^ atoms/cm^2^), reflection high-energy electron diffraction (RHEED) analysis showed diffraction rings associated with α-cristobalite and α-quartz, respectively. These results suggested that both Ne and Kr neutral atom irradiations caused a-to-c transformations, whereas the ion irradiations did not. The authors suggested that a greater extent of crystal growth may have occurred due to successive neutral irradiations. They also noted that since the two crystalline phases (α-cristobalite and α-quartz) were not found in the same sample, but rather one phase induced by Kr and one induced by Ne, the neutral beam irradiation conditions determine the pressure and temperature processes governing which crystalline phase will form. They also found that the preferential sputtering of O with the ionic counterparts (e.g., Kr^+^ and Ne^+^) prevented crystallinity changes due to the higher density of Si in the samples. They suggested that with neutral beam irradiation, local hotspots were formed by the bombardment of neutral atoms on the surface atoms.

Other work by Du et al. using 200 kV electron irradiation of amorphous SiO_2_ to a dose of 10^18^ C/m^2^ showed that rather than SiO_2_ crystallizing, Si crystallized in two steps [13]. Initially, in the amorphous SiO_2_ sample, amorphous Si formed and then eventually crystallized with continued electron irradiation. The authors stated that the bonds were broken by creating an O vacancy from the knock-on displacement as a result of energy transferred from the electron irradiation. Because of the O vacancies, the Si atoms then aggregated towards one another. They then argued that the beam heating decreased the nucleation barrier, while knock-on displacement increased the potential energy above the ground state, thus leading to the crystallization of Si.

### 3.5. Silicon Carbide

Silicon carbide is another ceramic material that has also demonstrated a-to-c transformations upon irradiation with Xe and Si ions. Amorphous SiC samples were prepared by Zhang et al. and then irradiated with 5 MeV Xe to a fluence of 1.15 × 10^16^ ions/cm^2^ at 700 K [54]. The amorphous film of SiC formed randomly oriented crystalline grains homogeneously distributed throughout the material, as shown in Figure 10a. While amorphous regions remain, the crystalline grains are of the 3C-SiC polymorph with a cubic structure consistent with the known zinc-blende structure of the 3C-SiC polymorph [64], as shown in Figure 10b. SAED patterns demonstrating the cubic crystalline structure formed are shown as an inset in Figure 10b. The crystalline grains contained stacking faults, identified by the shoulder peak at ~33.7° in the grazing incidence X-ray diffraction (GIXRD) spectrum and confirmed by TEM and Raman spectroscopy. The authors confirmed that the a-to-c transformation was irradiation-induced by conducting thermal annealing at 700 K and finding this was too low a thermal annealing temperature to induce crystalline grain nucleation and growth without irradiation.

The amorphous SiC to crystalline 3C-SiC transformation was also seen under 300 keV Si irradiation to a fluence of 1 × 10^17^ ions/cm^−2^ at 500 °C and 1050 °C by Heera et al. [55]. In their study, a series of samples were investigated at 500 °C, where one was thermally annealed, and the other was Si ion irradiated. The thermally annealed sample showed no changes from its original amorphous structure, whereas the ion irradiated sample demonstrated both epitaxial and polycrystalline crystallization. Although the ion irradiation-induced crystallization was not homogenous, neither epitaxial nor polycrystalline crystallization could be achieved at 500 °C through thermal annealing alone. The authors suggested that the polycrystalline layer consisted of two main zones, one of 6H-SiC and the other of 3C-SiC, resulting from ion beam-induced random nucleation and subsequent ion beam-enhanced grain growth. A series of samples were also studied at 1050 °C, and it was shown that the sample became fully crystalline following thermal annealing alone. The crystalline thermally annealed sample consisted of columns of hexagonal SiC, aligning with the underlying 6H substrate, though slightly inclined with respect to the substrate. The crystallinity under ion irradiation at 1050 °C was substantially improved, showing almost perfect epitaxial 6H-SiC with a near-surface columnar structured layer and aligned 3C-SiC grains. The overall number of 3C-SiC grains in the ion irradiated sample was smaller than the thermally annealed sample at elevated temperatures.

The effects of 690 keV Xe ion irradiation at dose levels of 10^15^, 5 × 10^15^, and 10^16^ ions/cm^2^ on amorphous MoSi_2_/SiC nanolayered composites were investigated by Lu et al. [56]. Initial electron diffraction showed that the alternating layers of MoSi_2_/SiC were amorphous. Following Xe^+^ irradiation, the layers lost their structural integrity, MoSi_2_ turned into a metastable hexagonal C40 phase, and the SiC began to spheroidize. This behavior occurred at all doses, although the changes were more pronounced with increasing dose. HRTEM showed that the C40 MoSi_2_ grains were nanocrystalline, while the SiC spheroidization indicated microstructural coarsening. These results from the irradiated sample indicated that the energy of the Xe ions enhanced diffusion similar to thermal annealing at 500 °C, although the irradiating particle energy is not sufficient to fully crystallize the SiC layer.

### 3.6. Silicates

Silicates are classified as any derivative of A_x_B_y_SiO_z_ excluding SiO and SiO_2_ and have exhibited a-to-c transformations under both electron and ion irradiation. Electron irradiation has been shown to induce compositional changes and crystallization in some silicates [2,57], while others exhibited epitaxial recrystallization [58]. Meldrum et al. studied the recrystallization of zircon (ZrSiO_4_) using two-stage irradiations [3]. First, amorphous ZrSiO_4_ was created by irradiating a natural crystalline zircon with 1.5 MeV Kr to a dose of 9 × 10^14^ ions/cm^2^ to ensure complete amorphization. Then, 200 keV electron irradiation at a beam current of 90 A/cm^2^ induced crystallization zones among the amorphous regions. Upon recrystallization, the diffraction patterns showed that the crystallites adopted cubic or tetragonal ZrO_2_ structures. When lowering the beam energy below 200 keV, zircon was unable to crystallize within a reasonable experimental time. The authors determined the maximum temperature increase due to beam heating to be only 19 °C, suggesting that thermal effects have little contribution to the observed crystallization [65].

While Meldrum’s work did not observe a compositional change associated with the amorphization or subsequent a-to-c transformation, Carrez et al. saw a compositional change followed by crystallization during 300 keV electron irradiation of MgSiO_3_ at room temperature [57]. Initial SAED showed diffuse rings indicative of amorphous samples, but as the fluence increased above 5 × 10^16^ electrons/cm^2^, small structures began forming on the surface of the sample too small for SAED identification, as shown in Figure 11. Above a fluence of 3 × 10^17^ electrons/cm^2,^ the material crystallized to MgO as indicated by new diffraction rings (Figure 11b). With increasing electron fluence, the rings became progressively sharper, indicating growth of the initial nuclei and newly formed crystallites. The newly formed structure was associated with periclase MgO, though as the fluence increased to 5 × 10^18^ electrons/cm^2^, the periclase rings disappeared, and new rings attributed to forsterite MgO appeared (Figure 11c). As the crystallites formed, so did the number of MgO-rich and MgO-poor domains. The authors suggested that SiO_2_ and MgO were not miscible under the experimental conditions, and the oxide separation is unsurprising as the phase diagram displayed a liquid immiscibility at high temperatures within the composition range of interest. They suggested that the sample went through two stages during irradiation. In the first stage, the sample was unstable and decomposed into MgO and SiO_2_; in the second stage, nanosized MgO crystallites formed. The nucleation and growth processes were activated by radiation-enhanced diffusion as a result of ionization processes as well as a thermodynamic driving force for crystallization. The paper did not report on the phase evolution of the SiO_2_ under irradiation.

Epitaxial recrystallization was observed by Bae et al. using two-step irradiation on an initially crystalline Sr_2_Nd_8_(SiO_4_)_6_O_2_. The amorphous Sr_2_Nd_8_(SiO_4_)_6_O_2_ was generated by irradiating the sample with 1.0 MeV Au to a fluence of 5 × 10^13^ ions/cm^2^ at 300 K, then subsequently irradiated with 200 keV electrons at a current of 0.29 A/cm^2^ at room temperature [58]. Following 840 s of exposure to an electron fluence of 1.53 × 10^21^ electrons/cm^2^, recrystallization was observed from both the amorphous/crystalline interface and surface, as shown in Figure 12. The SAED pattern in Figure 12a shows the initial amorphous diffraction pattern, and following electron irradiation, the SAED patterns in Figure 12d show distinct rings that correspond with recrystallization. In Figure 12d, the top right SAED pattern corresponds to the recrystallization occurring at the surface, and the bottom left SAED corresponds to the recrystallization at the interface. The recrystallization proceeded at the surface and the interface at roughly the same rates, with the recrystallized regions finally connecting in the center of the specimen after 3600 s of electron irradiation. This result demonstrated epitaxial growth from the electron beam irradiation due to the substrate acting as a template material on which recrystallization may occur. By utilizing the Bethe–Bloche equation, the authors calculated a maximum temperature increase of only 7 K from the electron beam. Further studies with thermal annealing were conducted, and the authors found no evidence of crystallization. Hence, they concluded that the recrystallization was an effect of electron irradiation alone. The authors stated that the ionization-induced processes caused by the electrons transferring their energy to the target atoms via inelastic interactions led to localized electronic excitations playing a role in the recrystallization. These excitations disturbed local atomic bonds and structures by lowering the energy barrier for defect recovery and recrystallization. Consequently, these electronic excitations lowered the energy barrier for defect recovery and recrystallization.

Other silicates which displayed crystallization as a result of Kr and Xe ion irradiation were ZrSiO_4_, ThSiO_4_, and HfSiO_4_, as reported by Meldrum et al. [2]. ThSiO_4_ and HfSiO_4_ were irradiated with 800 keV Kr ions, and ZrSiO_4_ was irradiated with 800 keV Xe ions. When ZrSiO_4_ was irradiated at temperatures above 900 K, the single crystal samples initially became amorphous and then, with increasing dose, decomposed into its component oxides: tetragonal ZrO_2_ and amorphous SiO_2_. This decomposition was similar to their previous results attained by electron irradiation [3], possibly demonstrating the immiscible properties of ZrO_2_ and SiO_2_. HfSiO_4_ similarly decomposed when heated to above 950 K into tetragonal HfO_2_ and amorphous SiO_2_. With ThSiO_4_, a similar irradiation-induced decomposition occurred at 1000 K into monoclinic ThSiO_4_ containing a network of distorted huttonite and randomly oriented ThO_2_. However, this decomposition process in ThSiO_4_ occurred with slower kinetics than the decompositions in ZrSiO_4_ and HfSiO_4_, and regions demonstrated more intense electron diffraction rings rather than rapid and complete crystallization. To determine whether the decomposition occurred due to thermal annealing alone, the ion beam amorphized samples were heated to 1100 K and held for 30 min without irradiation; neither crystallization nor decomposition occurred, so the authors concluded that the decomposition was associated with ion irradiation.

### 3.7. Other Ceramic Materials

Jiang et al. demonstrated that an a-to-c transformation could be induced in amorphous ZrC by irradiating with 30 keV He^+^ ions at an ion flux of 1.79 × 10^13^ ions/cm^2^-s to various fluences at 593 K [59]. The sample was prepared by depositing amorphous ZrC on a layer of crystalline ZrC using magnetron sputtering. At a fluence of 1.08 × 10^16^ ions/cm^2^, nanocrystals were visible in TEM cross-sections. At fluences ranging from 3.23 to 6.45 × 10^16^ ions/cm^2^, SAED analysis showed evidence of increased crystallization in the amorphous ZrC layer; i.e., diffuse rings corresponding to crystalline ZrC domains appeared at a fluence of 1.08 × 10^16^ ions/cm^2^ and the SAED patterns became more defined as the fluence increased. Additionally, irradiated samples were annealed at a temperature of 723 K to determine if heating at elevated temperatures would affect the morphology or crystallinity of irradiation-induced nanocrystals, but no significant changes were observed. The mechanism for the a-to-c transformation was primarily explained as a diffusion-assisted mechanism via the thermal spike model. Essentially, He^+^ irradiation transfers much of its energy to electrons in the target material and causes a sharp increase in temperature along the ion path, causing adjacent atoms to diffuse more rapidly. A related correlation between atomic distributions of Zr and C and the irradiation-induced crystallization was observed using EDS analysis of the sample cross-sections. This analysis found that the observed a-to-c transformation tended to occur at depths where the relative concentration of C was lower, and this concentration gradient occurred near the interface between the amorphous and crystalline layers of ZrC. Therefore, He^+^ irradiation seemed to preferentially form ZrC crystals near the amorphous-to-crystalline interphase where the C concentration was lower, and irradiation enhanced diffusion of Zr and C atoms would cause crystallites to gradually develop closer to the sample surface as the C to Zr ratio was decreased as a result of ion irradiation [59].

Allen and coworkers studied the effects of Kr ion irradiation on the crystallization of CoSi_2_ thin films [60]. The samples were partially crystallized from TEM electron beam irradiation operated at 300 kV prior to ion irradiation. Further crystallization was observed after 1.5 MeV Kr^+^ irradiation to a fluence of 8.5 × 10^14^ ions/cm^2^ at 300 K. The authors suggested that the prior electron irradiation led to a profuse number of crystallites with sizes that fell within the critical to supercritical size range that enabled the crystallites to continue to grow via subsequent ion irradiation-assisted recrystallization. The authors then irradiated another as-deposited film with 1.5 MeV Kr at a lower fluence of 3.4 × 10^14^ ions/cm^2^ at 300 K, then thermally crystallized the sample at 450 K. The irradiated portion of the sample showed higher crystallization rates, about double those of the non-irradiated portion of the sample. Additionally, when the ion dose was lower, specifically 3.4 × 10^13^ ions/cm^2^ at 300 K, the number density of crystallites was 30% lower, but the crystallite growth rates in the irradiated and non-irradiated portions of the sample were the same. This suggested that low-dose ion irradiation prior to thermal crystallization sufficiently induced critical or supercritical-sized crystallite nuclei that assisted in thermal crystallization.

Som et al. used two-step ion implantation and irradiation to induce recrystallization of an implanted Si_3_N_4_ layer in Si [61]. Initial implantation of 100 keV N ions into a Si wafer at 300 °C to a fluence of 8 × 10^17^ ions/cm^2^ created an amorphous Si_3_N_4_ layer while the Si above and below this layer remained crystalline. Subsequent irradiation with 100 MeV Ag ions was conducted to a fluence of 10^14^ ions/cm^2^. The implanted layer remained amorphous when the Ag ion irradiation was carried out at 150 °C, but the implanted layer crystallized when the Ag ion irradiation was carried out at 200 °C or higher. Using HRTEM, the *d*-spacing of the implanted then irradiation-crystallized layer matched with that of the hexagonal α-Si_2_N_4_ phase. This implanted layer *d-*spacing did not match that of the crystalline Si layers above and below it, so the authors suggested that the Ag ion irradiation caused the crystallization. The crystallization was explained by irradiation defect creation in the Si substrate. That is, as the Ag ions penetrated through the multi-layered structure and deposited in the substrate, they created vacancies and interstitials in the crystalline Si. These point defects were available at the amorphous/crystalline (a/c) interfaces between the layers and migrated through thermal diffusion. The vacancies in the amorphous layer enabled thermal vibrations of Si and N atoms to occur more freely and unconstrained, thus inducing redistribution that resulted in the crystallization of one monolayer at the a/c interface. These monolayers then progressed inward from both a/c interfaces bounding the amorphous layer.

Finally, Meldrum et al. used two-step irradiation to induce recrystallization of phosphates, namely ScPO_4_ and LaPO_4_ specimens [3]. The amorphous ScPO_4_ and LaPO_4_ were made by first irradiating single crystals with 1.5 MeV Kr to fluences of 4 × 10^14^ ions/cm^2^ and 2 × 10^15^ ions/cm^2^, respectively, to ensure complete amorphization. The amorphized specimens were subsequently irradiated with a 200 keV electron beam at a beam current of 1.45 A/cm^2^ at ambient temperature. The LaPO_4_ showed facile crystallization in mere seconds under the electron beam, whereas it took about 10 times as long for ScPO_4_ to fully recrystallize under the same electron beam conditions. When the electron beam current was increased, the crystallite size decreased for both phosphates, though crystallites in ScPO_4_ were consistently about 50% larger than those in LaPO_4_. When both materials were subjected to thermal annealing over a temperature range of 200–300 °C, solid phase epitaxy was achieved over a period of 1–3 h. To investigate the temperature increase as a result of electron beam heating, the Fisher model was utilized and showed that the temperature increased within the range of 4–40°C [65]. By contrast, temperatures in excess of 400 °C are needed to thermally anneal the amorphized samples for recrystallization.

## 4. Summary

The irradiation parameters and conditions leading to a-to-c transformations in ceramics appear to vary depending on the properties of the target material. While charged particle (i.e., electron and ion) irradiation has most readily induced a-to-c transformations in ZrO_2_ [40,48,54], TiO_2_ [42,43,44], and Al_2_O_3_ [45,46,47,50,51], neutral particles have also induced a-to-c transformations in SiO_2_ [12,13]_._ Attempting to discern a unifying explanation for these behaviors highlights the need to understand energy dissipation mechanisms in electron-electron, electron-ion, ion-ion, and atom-ion collisions. While the roles that nuclear and electronic stopping play in ceramic material evolution, and specifically in the a-to-c transformations, are not well-defined, there are some commonalities. For example, electron irradiation tends to require a minimum dose or dose rate to activate the a-to-c transformations [40,42,45,46]. Additionally, the rate of irradiation-induced crystallite nucleation and growth appears strongly dependent on the irradiation time, dose, and dose rate [40,42,45,46]. When the temperature is recorded and investigated as a variable, it is typically found that the a-to-c transformation occurs at temperatures considerably lower than the critical crystallization temperature for the same material under thermal annealing alone [40,42,45,46]. Additionally, the initiation of the a-to-c transformation and crystallization rate is influenced by the sample temperature [40,42].

The a-to-c transformation tendency appears somewhat insensitive to the irradiating particle energy. For example, irradiation-induced a-to-c transformations have been reported in ZrO_2_ using 2–35 keV Kr ions as well as 1 MeV Kr ions [41,48]. However, 1 MeV Kr ions do not always induce a-to-c transformation in ZrO_2_ [49]. Lastly, multiple authors have noted that initiation of the a-to-c transformation can be assisted by the presence of an adjacent crystalline layer and directed via epitaxial growth [47,50,51]. Beyond these commonalities and minor trends, the archival literature provides insight into the purported mechanisms of irradiation-induced a-to-c transformations, as well as implications of a-to-c transformations on material structure and properties. These aspects will be discussed in this section.

### 4.1. Proposed A-to-C Transformation Mechanisms

Irradiation-assisted crystalline-to-amorphous (c-to-a) transformations in ceramic materials have been described by several models, recently summarized by Weber [25]. These models describe irradiation-induced amorphization as dependent on nuclear and ionization processes, thermal heating and recovery, and kinetic processes. Through these dependencies, amorphization relies on parameters such as defect concentration, damage cross-section, ion dose or fluence, and temperature [25]. The opposite a-to-c process is thought to rely on the same parameters, but the specific interplay between them that leads to a-to-c transformations is less understood.

Naguib and Kelly presented three possible mechanisms to explain their experimental observations of irradiation-induced a-to-c transformations in ZrO_2_ films: (a) displacement spikes [66], (b) thermal spike-induced heating, and (c) radiation-enhanced diffusion [48]. The displacement spike model would essentially assume that displaced atoms in an amorphous matrix will rearrange into crystal structures with no activation energy requirement [48,66]. The authors found this model to be unlikely on the basis that their own experiments (summarized in Section 3.1) showed that crystallization is energy- and dose rate-dependent, and it fails to explain why similar metal oxides do not undergo a-to-c transformations under irradiation [48,67].

With regard to the thermal spike mechanism, the effectiveness of a thermal spike heating process is [48,68]:(2)$$\left(D_{c}^{o}\frac{t}{\lambda }\right)e^{\left(\frac{-\Delta H_{c}}{RT}\right)}>\lambda$$
where $D_{c}^{o}$ is the pre-exponential diffusion coefficient for crystallization, λ is the average atomic spacing (~2 Å), $\Delta H_{c}$ is the activation enthalpy, R is the ideal gas constant, and T is the temperature. For the thermal spike-induced heating model to be valid, one must assume that crystallization will occur if the term ($D_{c}^{o}t/\lambda$) is significantly larger than λ. However, Naguib and Kelly calculated a value of 1 Å for ($D_{c}^{o}t/\lambda$), which is lower than λ and implies thermal spikes played no role in their observed a-to-c transformation in ZrO_2_. However, due to the uncertainty in the actual values of $D_{c}^{o}$, t, and λ, the model in Equation (2) could not be entirely ruled out.

Naguib and Kelly’s final proposed mechanism, enhanced radiation diffusion, assumes that crystal growth occurs due to impacts with pre-existing crystallites in the amorphous matrix. If such a radiation-enhanced diffusion mechanism is combined with assumptions from the thermal spike-induced heating model, the calculated value for ($D_{c}^{o}t/\lambda$) becomes ~4 Å, which satisfies the condition for significant crystallization in Equation (2). Nevertheless, the thermal spike cannot fully explain the a-to-c transformations reported to occur well below the crystallization temperatures for the observed phase change.

Using their study of electron irradiation-induced crystallization in amorphous Fe_85_B_15_ alloy as a model system, Qin et al. proposed a non-equilibrium thermodynamically driven mechanism, shown schematically in Figure 13. In this model, they argue that the nucleation theory of minimized Gibbs free energy is not applicable in irradiation-induced a-to-c transformations since these transformations occur below the crystallization temperature of materials and, therefore, cannot be attributed to thermal heating by ion or electron beams [69].

Instead, they propose that the initial amorphous phase is a high-energy metastable state and that crystallization represents a lower energy state that may be reached due to atomic rearrangement prompted by irradiation. Whether the a-to-c rearrangement proceeds depends on the net energy balance. The energy introduced by irradiation (∆*E*_n_) is the sum of two components: the energy required to create stored point defects (∆*E*_sto_) and the energy used to stimulate the original amorphous state to a more disordered state (∆*E*_dis_), such that ∆*E*_n_ = ∆*E*_sto_ + ∆*E*_dis_. Subsequently, as the a-to-c transformation proceeds, the energy released during atomic rearrangement (∆*E*_r_) is the sum of ∆*E*_dis_ (which now represents the energy dissipated by atomic rearrangements from the stimulated state) and the free-energy difference between the original amorphous state and the final crystalline state (∆*E*_rea_), such that ∆*E*_r_ = ∆*E*_dis_ + ∆*E*_rea_. The net energy balance for a-to-c transformation to proceed thus requires ∆*E*_r_ > ∆*E*_n_ or ∆*E*_rea_ > ∆*E*_sto_. In other words, ceramics only undergo an irradiation-induced a-to-c transformation when the free energy difference between the amorphous and crystalline states exceeds the energy stored by creating irradiation-induced point defects.

Alternatively, Meldrum et al. attributed a-to-c transformations to a nucleation and growth mechanism related to ionization processes and the rearrangement of bonds in the amorphous matrix [3]. Additional parameters such as structural rigidity, irradiation-induced bond hybridization, and bond breaking/reformation at amorphous-crystalline interfaces play a critical role in the a-to-c transformation mechanism [3,23,48,49,70]. For example, in a study of electron beam-induced recrystallization of LaPO_4_, ScPO_4_, and ZrSiO_4_, the amorphous phosphates crystallize at higher rates than ZrSiO_4_ due to their less rigid crystal structures [3]. Furthermore, LaPO_4_ exhibits rapid crystallization compared to ScPO_4_ due to lower coordination of the PO_4_ tetrahedrons in the LaPO_4_ crystal structure [3].

### 4.2. Morphological and Functional Evolution Due to A-to-C Transformation

Irradiation-induced a-to-c transformations can lead to morphological changes as well as changes to physical properties and functionality. Much of the literature on a-to-c transformations in ceramic materials summarized in this perspective focuses primarily on the structural and phase evolution under irradiation. However, several of these studies have also reported some notable morphological changes, including swelling, formation of polycrystalline and nanocrystalline substructures, and physical distortion of target materials. Others have reported on the implications of a-to-c transformations on dielectric properties and electrochemical performance. These morphological and functional changes will be discussed in this section.

In their TEM in situ irradiation study, Yang et al. showed that irradiation-induced a-to-c transformations in TiO_2_ nanotubes resulted in gross morphological changes, specifically bending and curling of nanotubes. Partial crystallization of domains in the side walls of the nanotubes caused internal stresses that are relieved by mechanical bending [44]. This conclusion was supported both by the observation of partial crystallization in the bent tubes (recall Figure 7a) and by molecular dynamics volume calculations which show that significant densification is a predicted outcome of an amorphous-to-anatase phase transformation [44]. Morphological volumetric swelling also occurred in amorphous SiC under Si ion irradiation [10].

Functional electrochemical evolution associated with a-to-c transformations has been reported by Smith et al. in their study of the effects of proton irradiation on TiO_2_ nanotubes for Li-ion battery negative electrodes [43]. They found that irradiation of amorphous TiO_2_ nanotubes at 250 °C resulted in an a-to-c transformation to mainly disordered rutile TiO_2,_ while irradiation at room temperature resulted in a mixed domain of rutile and anatase crystallites amongst amorphous phases. This phase evolution enhanced the capacity of the 250 °C irradiated sample to about 240 mAh/g compared to about 200 mAh/g for an unirradiated anatase control. This improvement was attributed to the irradiation-induced a-to-c transformation and associated irradiation-induced defects. However, the room temperature irradiated sample had a capacity of about 130 mAh/g; this diminished performance compared to the control was attributed to the mixed anatase, rutile, and amorphous domains negatively impacting ion mobility and diffusion rates of charge carriers in the Li-ion battery [43].

Mizutani et al. [12] saw a 5 nm thick crystalline layer formed in amorphous SiO_2_ following neutral Ne atom irradiation at 350 eV to a fluence of 10^17^ atoms/cm^2^. TEM and RHEED results indicated the formation of platelet-shaped SiO_2_ microcrystals ~1 nm in thickness with an α-cristobalite structure. Microcrystals were also formed by neutral Kr atom irradiation of the same amorphous SiO_2_ at the same irradiating particle energy of 350 eV to the same fluence of 10^17^ atoms/cm^2^. However, the microcrystals had an α-quartz structure. Regardless of the irradiation-induced microcrystal structure, both the Ne and Kr irradiated samples exhibited a 15% increase in dielectric constant over the unirradiated amorphous SiO_2_.

## 5. Commentary & Outlook

Disorder-order transformations in irradiated materials are an unusual yet rather ubiquitous phenomenon. These transformations are particularly challenging to observe, simulate, and ultimately understand because they entail electron-ion and electron-electron interactions that occur on extremely short timescales (~femtoseconds to picoseconds) and length scales (~sub-Å to Å). Both experimental and computational techniques have limited sensitivity, resolution, accuracy, and efficiency that precludes meaningful measurements and simulations at these length and time scales. Moreover, the irradiation-induced a-to-c phenomenon is remarkably complex, being influenced by irradiation damage physics, mass and thermal transport phenomena, and microstructural evolution with the formidable subtleties of material defect chemistry.

Understanding energy dissipation during irradiation may be crucial for unlocking the a-to-c transformation mechanisms. Several studies have suggested that irradiation-induced ordering can only proceed when electronic energy losses (i.e., ionization effects) occur, while nuclear energy losses are not necessary for irradiation-induced ordering. For example, in an ilmenite-hematite (FeTiO_3_-Fe_2_O_3_) solid solution, chemical ordering occurs during ~MeV proton irradiation and is attributed to localized annealing effects due to electronic energy deposition [71]. However, the same ilmenite-hematite solid solution material is resistant to ordering under high energy (~100 s of MeV) neutron irradiation, possibly due to the nuclear nature of the energy losses [72]. Huang et al. [73] also investigated the relative contributions of electronic and nuclear energy losses in amorphous Gd_2_Zr_2_O_7_. They observed local nanoscale crystallization to the fluorite phase following 120 keV electron irradiation at 400 °C, where ionization dominates energy losses. However, the specimen remained amorphous following 2 MeV electron irradiation, at which both displacement and ionization damage occurred.

At the same time, several studies have reported an absence of a-to-c transformation in amorphous SiOC irradiated with 100 keV He^+^ ions at room temperature and 600 °C to doses as high as 20 displacements per atom (dpa) [6], and in nanolayered amorphous SiOC/Fe structures [74,75]. Irradiation temperatures, doses, and particles were not remarkably different in these works compared to the studies summarized in Section 3 of the present article, wherein a-to-c transformations were observed. Hence, there may not be simple or universally applicable criteria for a-to-c transformations. Rather, the irradiation-induced or irradiation-assisted a-to-c transformation mechanisms may be controlled by a tremendous number of irradiation parameters and material properties in an intricately coupled manner that remains poorly understood.

The irradiation-induced morphological instabilities of the TiO_2_ nanotubes exhibit similar characteristics to the irradiation-induced morphological instabilities previously reported in germanium under electron [76] and ion [66] irradiation. While germanium and other semiconductors are excluded from this perspective (since this perspective instead focuses specifically on ceramic materials), synergies between ceramics and semiconductors are worth noting, given their distinct differences in bonding type and crystal structure. This underscores the need for researchers to understand the fundamental physics of radiation interactions in the matter at an electronic level in order to control and tailor phase transformability under irradiation.

Systematic investigations are necessary to begin to define the envelope of environmental conditions and material properties within which irradiation can induce a-to-c transformations. These investigations must take advantage of state-of-the-art developments in characterization and simulation techniques that now enable us to probe rapid timescale, small-length scale materials physics. Such studies are necessary to establish unifying explanations for the a-to-c transformation mechanisms across the broad class of ceramic materials. Attaining such a mechanistic understanding of the a-to-c phenomenon is key to harnessing and tailoring the unique properties and functionalities of amorphous ceramics for current and future applications in extreme irradiation environments.

## Figures and Tables

**Figure 1 materials-15-05924-f001:**
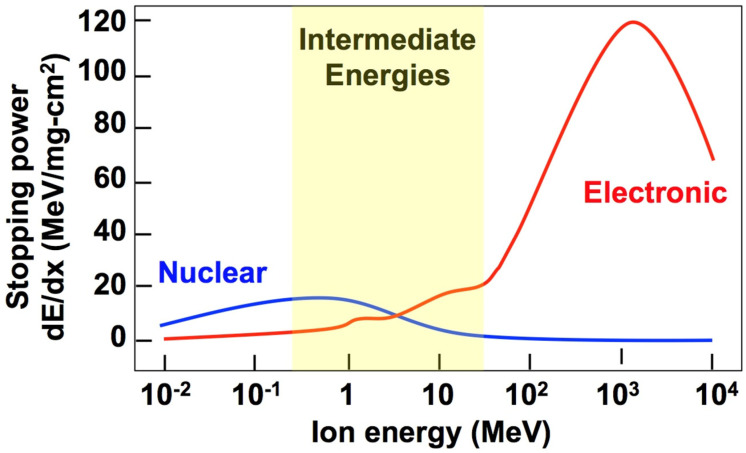
Illustration of three relevant regimes of stopping power, where electronic energy losses are dominant at high irradiating particle energies, nuclear energy losses are dominant at low energies, and both electronic and nuclear energy losses contribute at intermediate energies.

**Figure 2 materials-15-05924-f002:**
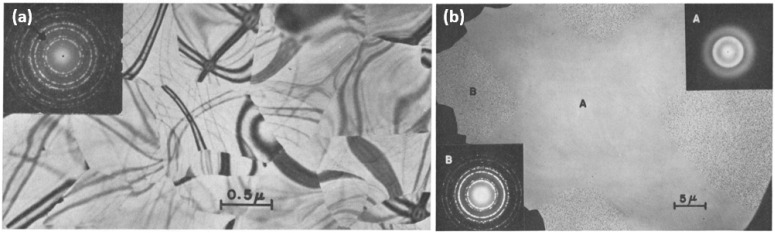
TEM images and inset SAED patterns of (**a**) fully crystallized ZrO_2_ produced by TEM in situ pulse heating; (**b**) amorphous ZrO_2_ film irradiated with 20 keV Kr ions to a fluence of 9 × 10^15^ ions/cm^2^ through a 400-mesh Cu grid with a 5 µA/cm^2^ flux; the region shielded by the mesh (inset image marked ‘A’) remained amorphous, while the unshielded region bombarded by ions (inset image marked ‘B’) underwent an a-to-c transformation. Reproduced with permission from ref. [48].

**Figure 3 materials-15-05924-f003:**
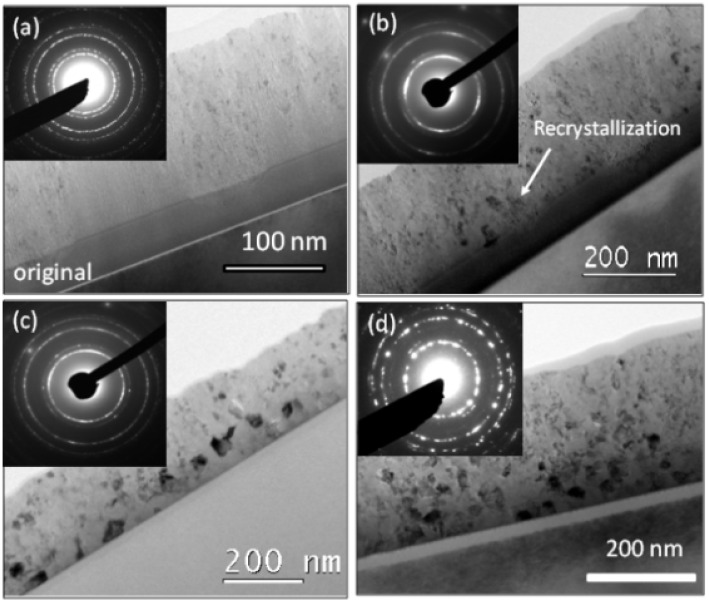
In situ TEM images with SAED insets of bilayer films of nanocrystalline cubic and amorphous ZrO_2_, showing the crystal and microstructure evolution at ion fluences of (**a**) 0, original sample, (**b**) 3.13 × 10^14^ ions/cm^2^, (**c**) 1.88 × 10^15^ ions/cm^2^, and (**d**) 3.13 × 10^15^ ions/cm^2^. Reproduced with permission from ref. [41].

**Figure 4 materials-15-05924-f004:**
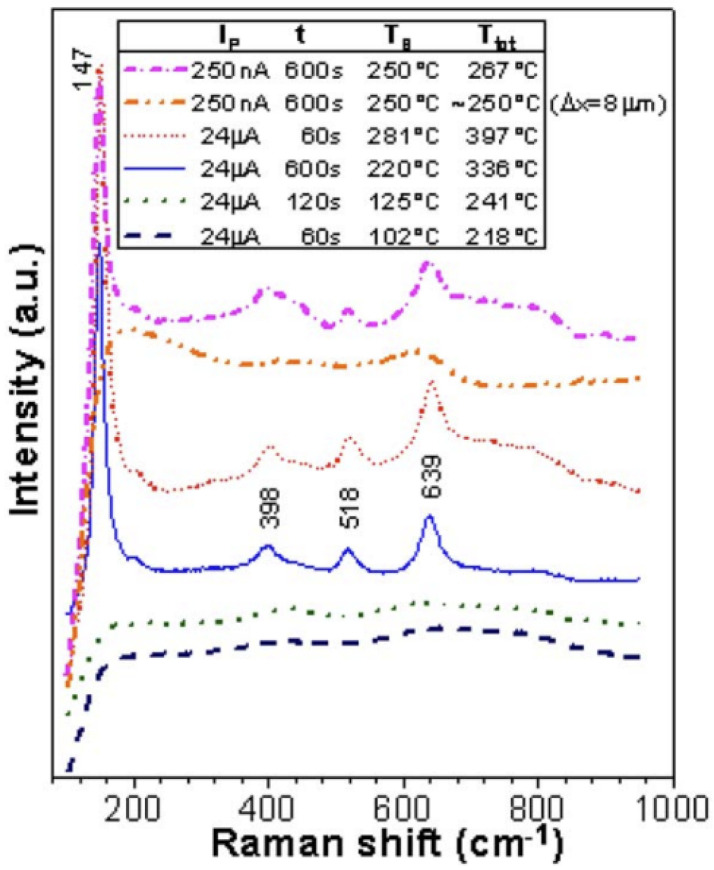
Raman spectra for TiO_2_ films after SEM electron irradiation under various beam current, temperature, and time conditions. Reproduced from ref. [42].

**Figure 5 materials-15-05924-f005:**
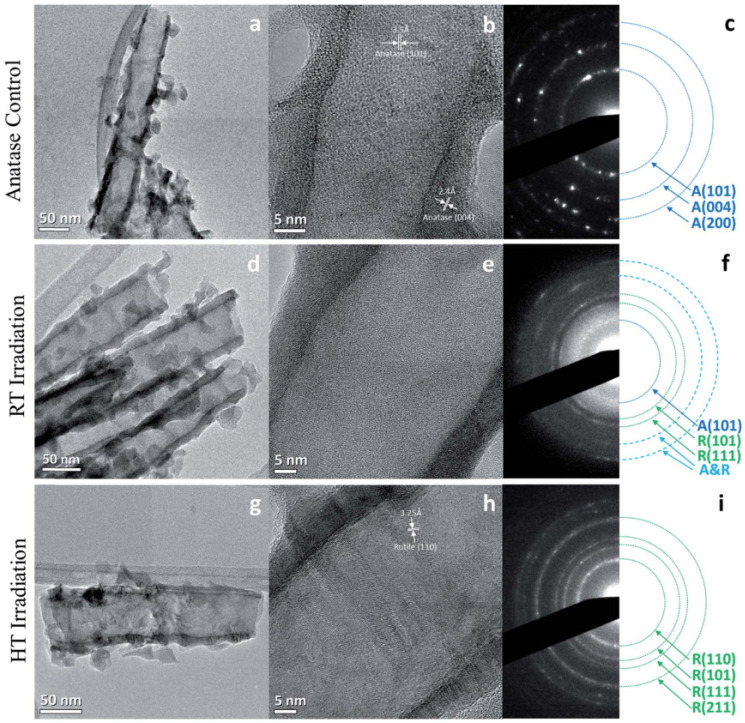
Low magnification TEM images of (**a**) unirradiated anatase TiO_2_ nanotubes, (**d**) RT proton irradiated nanotubes, and (**g**) 250°C (HT) irradiated nanotubes, showing retained structural morphology after irradiation. HRTEM images of (**b**) unirradiated anatase TiO_2_ nanotubes, (**e**) RT proton irradiated nanotubes, and (**h**) 250°C (HT) irradiated nanotubes and their corresponding SAED patterns (**c**, **f**, and **i**, respectively). Reproduced with permission from ref. [43].

**Figure 6 materials-15-05924-f006:**
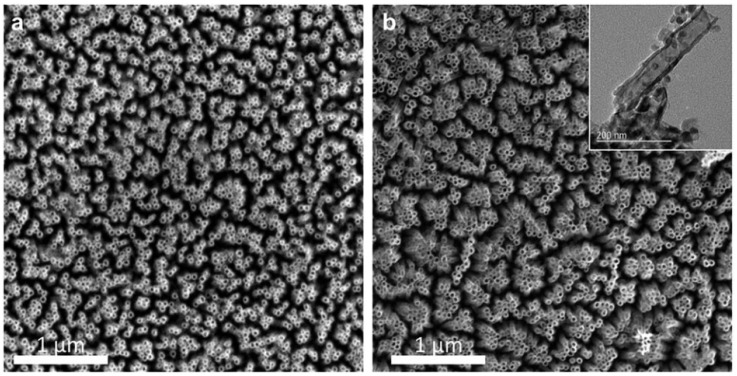
SEM images of TiO_2_ nanotube film (**a**) before and (**b**) after proton irradiation with inset TEM image, showing retained structural morphology. Reproduced with permission from ref. [43].

**Figure 7 materials-15-05924-f007:**
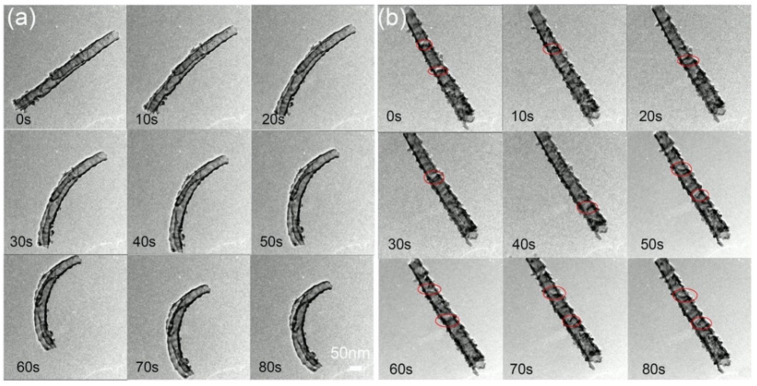
TEM images acquired at 10 s intervals during in situ 46 keV Au^−^ irradiation up to a total irradiation time of 80 s of (**a**) amorphous and (**b**) anatase TiO_2_ nanotubes. Reproduced with permission from ref. [44].

**Figure 8 materials-15-05924-f008:**
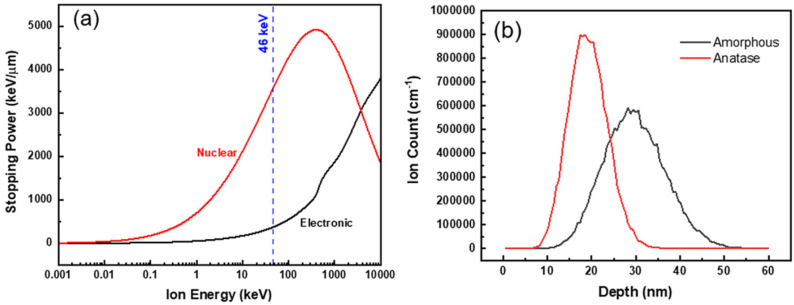
(**a**) Nuclear and electronic stopping power profiles and (**b**) ion range profiles for amorphous TiO_2_ nanotubes. Reproduced with permission from ref. [44].

**Figure 9 materials-15-05924-f009:**
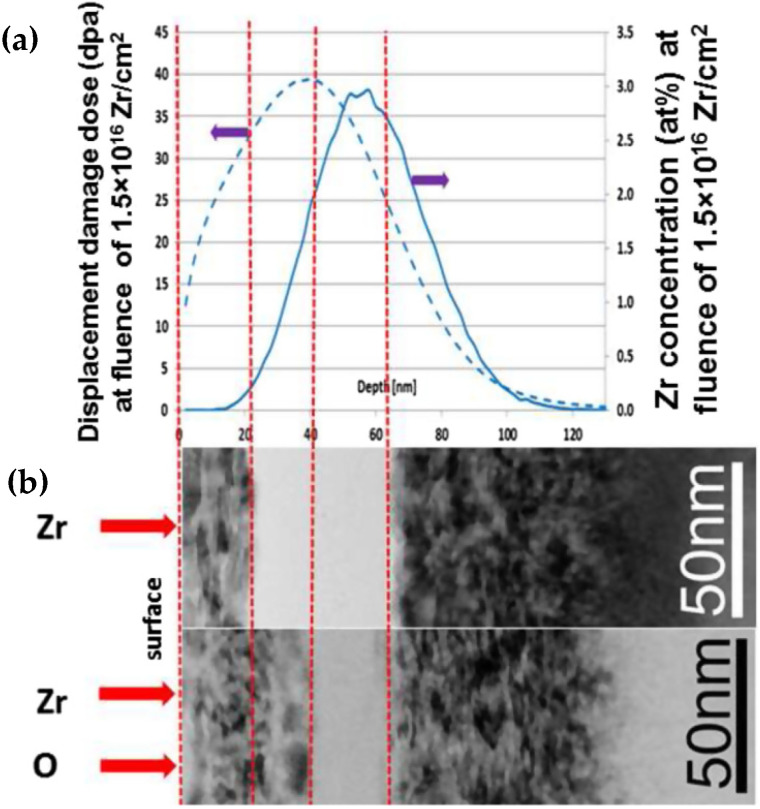
Cross-sectional TEM images of α-Al_2_O_3_ (**a**) irradiated with Zr^+^ ions to 1.5 × 10^16^ ions/cm^2^ and subsequently with (**b**) O^+^ ions to 2.3 × 10^16^ ions/cm^2^, demonstrating the extent of recrystallization of the buried amorphous layer. Overlaid SRIM damage (blue dotted line) and ion range (solid blue line) profiles show a clear relationship between the observed phase structures and the irradiation damage and Zr^+^ ion implantation peaks (red dotted lines). Reproduced with permission from ref. [51].

**Figure 10 materials-15-05924-f010:**
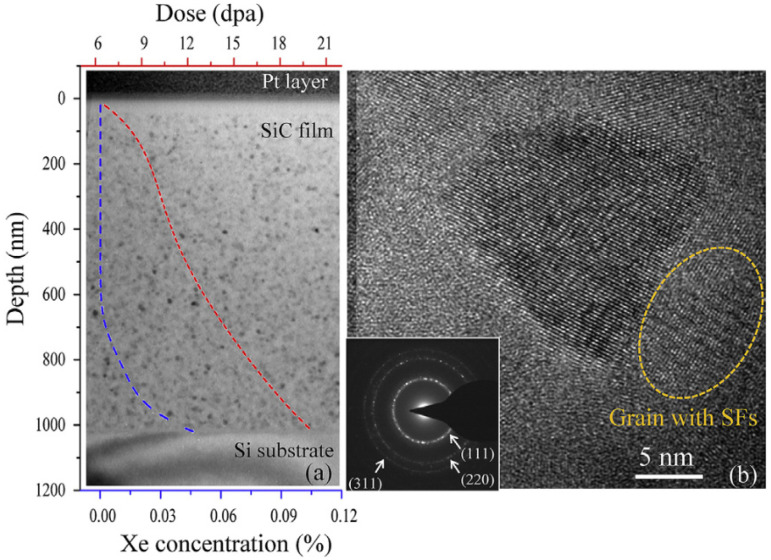
(**a**) TEM images of amorphous SiC film irradiated with 5 MeV Xe ions at 700 K to a fluence of 1.15 × 10^16^ ions/cm^2^, with overlaid ion implantation (blue dotted line) and dose profiles (red dotted line); (**b**) HR-TEM images and inset SAED showing cubic-structured crystallite formation. Reproduced with permission from ref. [54].

**Figure 11 materials-15-05924-f011:**
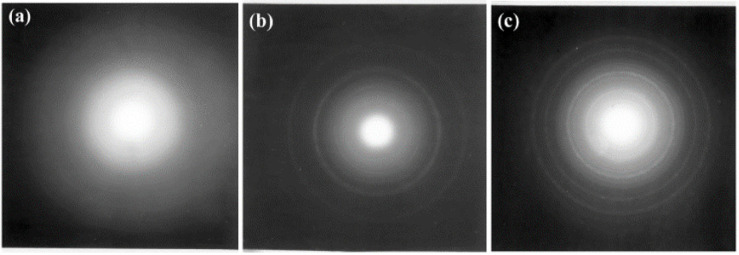
SAED patterns of amorphous MgO at fluences of (**a**) 0, i.e., before irradiation, (**b**) 3 × 10^17^ electrons/cm^2^, showing rings indicating crystallization, and (**c**) 5 × 10^18^ electrons/cm^2^, where MgO rings disappeared and new forsterite rings appeared. Reproduced from ref. [57].

**Figure 12 materials-15-05924-f012:**
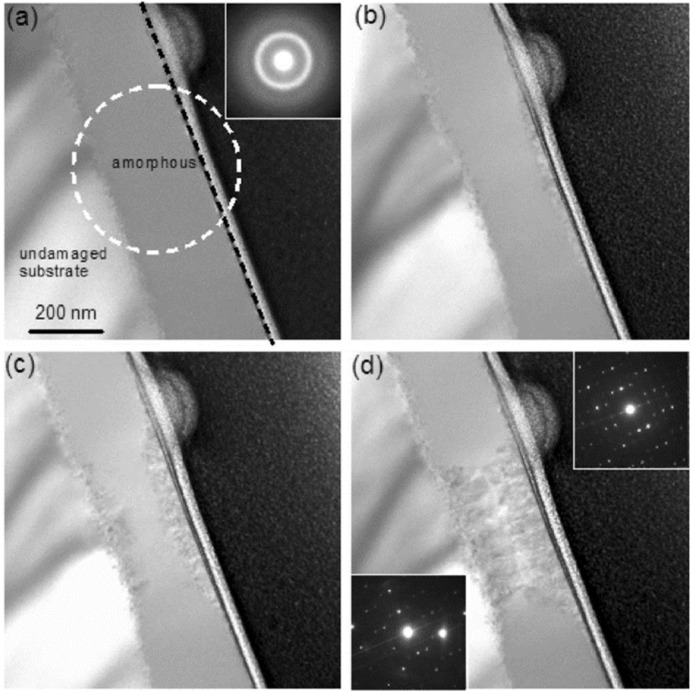
TEM images showing electron irradiation-induced structural changes of Sr_2_Nd_8_(SiO_4_)_6_O_2_ at the surface and the amorphous/crystalline interface at fluences of (**a**) 0, i.e., before irradiation, (**b**) 1.53 × 10^21^ electrons/cm^2^, (**c**) 3.60 × 10^21^ electrons/cm^2^, and (**d**) 6.55 × 10^21^ electrons/cm^2^. Reproduced with permission from ref. [58].

**Figure 13 materials-15-05924-f013:**
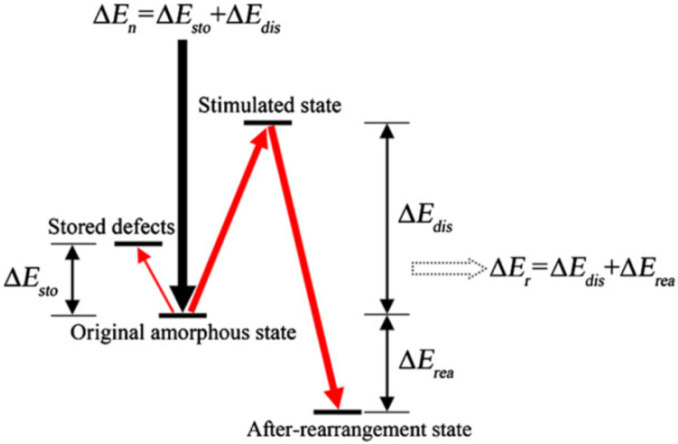
Schematic showing a proposed thermodynamic driven mechanism for the a-to-c transformations in amorphous materials. Reproduced with permission from ref. [69].

**Table 1 materials-15-05924-t001:** Summary of literature reporting irradiation-induced a-to-c transformations in metal oxides; AF = amorphous as fabricated, IA = amorphized by irradiation prior to second irradiation.

Target Material	Initial State of Target	Irradiating Particle	Energy (keV)	Fluence (ions/cm^2^)	Temp. (°C)	Irradiation-Induced Ordering	Ref.
ZrO_2_	AF	e^−^	200, 400 kV	–	20, −268	Crystallization	[40]
ZrO_2_	IA	Kr	2–35	~7 × 10^14^–2× 10^16^	20	25–75 nm nanocrystals	[48]
ZrO_2_	AF	Kr	1000	Various	–	Crystallization from amorph domains	[41]
TiO_2_	AF	e^−^	20 kV	–	20, 250	Crystallization	[42]
TiO_2_	AF	H^+^	200	2.18 × 10^17^	20, 250	Crystallization	[43]
TiO_2_	AF	Au^−^	46	2.3 × 10^14^	20	Nanocrystals	[44]
Al_2_O_3_	AF	e^−^	100 kV	–	–	Crystallization	[45,46]
Al_2_O_3_	IA	In, Si	100, 1500	2.7 × 10^16^, 3 × 10^16^	400	Formation of epitaxial γ-Al_2_O_3_	[50]
Al_2_O_3_	AF	Ar, O	360, 180	–	400, 500, 600	Formation of epitaxial γ-Al_2_O_3_	[47]
Al_2_O_3_	IA	Zr,O	175, 55	–	20	Epitaxial recrystallization	[51]
SiO	AF	Ni, Pb	575,000, 863,000	1 × 10^11^, 1× 10^13^	20	Si nanocrystals	[52]
SiO	AF	He	80	7 × 10^20^	~850	Si nanocrystals	[53]
SiO_2_	AF	Ne, Kr	400, 350	2 × 10^17^, 1 × 10^17^	–	Crystallization	[12]
SiO_2_	AF	e^−^	200 kV	–	–	Crystallization	[13]
SiC	AF	Xe, Si	5000	1.15 × 10^16^	~426	Crystallization	[54]
SiC	IA	Si	300	1e^17^	500, 1050	Crystallization	[55]
MoSi_2_/SiC	AF	Xe	690	1 × 10^15^, 5 × 10^15^, 1 × 10^16^	-	Crystallization	[56]
ZrSiO_4_	IA	e^−^	200	-	-	Crystallization zones	[3]
MgSiO_3_	AF	e^−^	300	5 × 10^16^ e^-^/cm^2^	-	Crystallization	[57]
Sr_2_Nd_8_(SiO_4_)_6_O_2_	IA	e^−^	200	1.53 × 10^21^ e^-^/cm^2^	20	Crystallization	[58]
ZrSiO_4_	IA	Xe	800	-	~626	Tetragonal ZrO_2_ and amorphous SiO_2_	[2]
ThSiO_4_	IA	Kr	800	-	~676	Monoclinic ThSiO_4_ and distorted huttonite/randomly oriented ThO_2_	[2]
HfSiO_4_	IA	Kr	800	-	~726	Tetragonal HfO_2_ and amorphous SiO_2_	[2]
ZrC	AF	He	30	1.08–6.45 × 10^16^	~320	Crystallization	[59]
CoSi_2_	AF	Kr	1500	8.5 × 10^14^	~26	Crystallization	[60]
Si_3_N_4_	AF	Ag	100,000	1 × 10^14^	200	Crystallization	[61]
ScPO_4_ and LaPO_4_	IA	e^−^	200	-	-	Crystallization	[3]

## Data Availability

Not applicable.

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
