# Peer review of "Irradiation-Induced Amorphous-to-Crystalline Phase Transformations in Ceramic Materials"

_materials, 2022, doi:10.3390/ma15175924_

Round 1
Reviewer 1 Report
The paper reviews the literature regarding the a-to-c transformation of ceramics under irradiation. The review work is comprehensive, and provides insights for this interesting yet not significantly noticeable topic in the nuclear materials field. In general, this is a great summary of this topic, and I’d be happy to add this paper into my reading list. I just have some questions, more than comments. See them below:
1. The formation of nanotube in TiO2 is very similar to some of the work done on irradiated germanium
2. In Figure 12(d), what are the two diffraction patterns corresponding to?
3. Have authors looked into metal-nitride materials (for example, ZrN, UN, etc.)? These materials are drawing more attention in recent years.
Author Response
The paper reviews the literature regarding the a-to-c transformation of ceramics under irradiation. The review work is comprehensive, and provides insights for this interesting yet not significantly noticeable topic in the nuclear materials field. In general, this is a great summary of this topic, and I’d be happy to add this paper into my reading list. I just have some questions, more than comments. See them below:
Response: The authors appreciate that the reviewer found the manuscript comprehensive and insightful. We have made changes to the manuscript to address the comments accordingly.
- Comment: The formation of nanotube in TiO2 is very similar to some of the work done on irradiated germanium.
Response: Thanks for the comment. It is interesting that TiO2 nanotubes show similar behavior of irradiated germanium. We have added the following text to the manuscript to refer to this similarity:
The irradiation-induced morphological instabilities of the TiO2 nanotubes exhibit similar characteristics to the irradiation-induced morphological instabilities previously reported in germanium under electron [76] and ion [66] irradiation. While germanium and other semiconductors are excluded from this perspective (since this perspective instead focuses specifically on ceramic materials), synergies between ceramics and semiconductors are worth noting given their distinct differences in bonding type and crystal structure. This underscores the need for researchers to understand the fundamental physics of radiation interactions in the matter at an electronic level in order to control and tailor phase transformability under irradiation.
- Comment: In Figure 12(d), what are the two diffraction patterns corresponding to?
Response: Thank you for the comment, we agree that the SAED patterns need to be explained better. The following excerpt was included in the manuscript to explain Figure 12 better:
…Following 840 sec of exposure to an electron fluence of 1.53x1021 electrons/cm2, recrystallization was observed from both the amorphous/crystalline interface and surface, Figure 12. The SAED pattern in Figure 12a shows the initial amorphous diffraction pattern, and following electron irradiation, the SAED patterns in Figure 12d shows distinct rings that correspond with recrystallization. In Figure 12d, the top right SAED pattern corresponds to the recrystallization occurring at the surface and the bottom left SAED corresponds to the recrystallization at the interface…
- Comment: Have authors looked into metal-nitride materials (for example, ZrN, UN, etc.)? These materials are drawing more attention in recent years.
Response: Thank you for the suggestion. We have not found any further published work showing irradiation-induced amorphous-to-crystalline transformations in metal-nitride materials, including ZrN and UN which are both studied almost exclusively in the initially crystalline state. But while conducting this further literature review, we found an additional study showing irradiation-induced a-to-c transformation in ZrC, which has been added to the manuscript:
Jiang et al. demonstrated that an a-to-c transformation could be induced in amorphous ZrC by irradiating with 30 keV He+ ions at an ion flux of 1.79x1013 ions/cm2-s to various fluences at 593 K [59]. The sample was prepared by depositing amorphous ZrC on a layer of crystalline ZrC using magnetron sputtering. At a fluence of 1.08x1016 ions/cm2, nanocrystals were visible in TEM cross-sections. At fluences ranging from 3.23 to 6.45x1016, SAED analysis showed evidence of increased crystallization in the amorphous ZrC layer. i.e., diffuse rings corresponding to crystalline ZrC domains appeared at a fluence of 1.08x1016 ion/cm2 and the SAED patterns became more defined at the fluence increased. Additionally, irradiated samples were annealed at a temperature of 723 K to determine if heating at elevated temperatures would affect the morphology or crystallinity of irradiation-induced nanocrystals, but no significant changes were observed. The mechanism for the a-to-c transformation was primarily explained as a diffusion-assisted mechanism via the thermal spike model. Essentially, He+ irradiation transfers much of its energy to electrons in the target material, and causes a sharp increase in temperature along the ion path – causing adjacent atoms to diffuse more rapidly. A related correlation between atomic distributions of Zr and C and the irradiation induced crystallization was observed using EDS analysis of the sample cross-sections. This analysis found that the observed a-to-c transformation tended to occur at depths where the relative concentration of C was lower, and this concentration gradient occurred near the interface between the amorphous and crystalline layers of ZrC. Therefore, He+ irradiation seemed to preferentially form ZrC crystals near the amorphous-to-crystalline interphase where the C concentration was lower, and irradiation enhanced diffusion of Zr and C atoms would cause crystallites to gradually develop closer to the sample surface as the C to Zr ratio was decreased as a result of ion irradiation [59].
Reviewer 2 Report
Authors of the paper “Review of irradiation-induced amorphous-to-crystalline phase transformations in ceramic materials” perform a full and timely review of amorphous ceramics and ways to obtain it in different conditions.
The review provides sources of both original and latest references on the topic.
However, I recommend a minor revision of the paper.
Minor mistakes and typos are everywhere in the manuscript text: extra gaps, missing gaps, references outside the sentences dots, commas instead of dots.
I have a question about the Funding source: it is “National Science Foundation”. But of what nation is it? The Journal is international, so readers from all countries will read the paper.
Author Response
Authors of the paper “Review of irradiation-induced amorphous-to-crystalline phase transformations in ceramic materials” perform a full and timely review of amorphous ceramics and ways to obtain it in different conditions. The review provides sources of both original and latest references on the topic. However, I recommend a minor revision of the paper.
- Comment: Minor mistakes and typos are everywhere in the manuscript text: extra gaps, missing gaps, references outside the sentences dots, commas instead of dots.
Response: The authors appreciate the reviewer has pointed these out and we have accordingly addressed these issues.
- Comment: I have a question about the Funding source: it is “National Science Foundation”. But of what nation is it? The Journal is international, so readers from all countries will read the paper.
Response: Thank you for the comment. We have updated the funding source in the revised manuscript as “U.S. National Science Foundation”.
Reviewer 3 Report
Although the researchers have conducted a comprehensive review in the area of Irradiation-Induced Amorphous-to-Crystalline Phase Transformations, it needs to be improved in the light of the following observations:
1. The authors are recommended to use technical terms in keywords list.
2. The authors should incorporate a strong conclusion(s) section to summarize and correlate research findings of the literature they have included in this review. It can include the different sets of parameters investigated by researchers and how do their findings agree with one another.
3. A formal technical reporting style should be adhered to and first person referral should be avoided.
4. A list of nomenclature should be included in the article before the introduction section.
5. SAED patterns should be discussed and compared in detail. The authors need to expand on the changes in the patterns at different irradiation exposures?
6. The article could be expanded to include more research articles for a more comprehensive review.
Author Response
Although the researchers have conducted a comprehensive review in the area of Irradiation-Induced Amorphous-to-Crystalline Phase Transformations, it needs to be improved in the light of the following observations:
- Comment: The authors are recommended to use technical terms in keywords list.
Response: We feel that the keywords are indeed technical terms.
- Comment: The authors should incorporate a strong conclusion(s) section to summarize and correlate research findings of the literature they have included in this review. It can include the different sets of parameters investigated by researchers and how do their findings agree with one another.
Response: The content the reviewer is requesting (summary of findings, parameters studied in the literature, agreement between the literature) has been concisely summarized in the “Commentary and Outlook” Section. Since this is a perspective style article (rather than original research), we do not feel it appropriate to make a Conclusions Section, but rather conclude with our Commentary and Outlook.
- Comment: A formal technical reporting style should be adhered to and first person referral should be avoided.
Response: We agree with the reviewer that this work should only contain a formal technical reporting style and all forms of first-person referral have been removed.
- Comment: A list of nomenclature should be included in the article before the introduction section.
Response: Thank you for the comment. The journal does not require a list of nomenclature and the authors followed the Instructions for Authors by the journal and believe the existing terms and nomenclature used in the manuscript are adequately defined in the present draft.
- Comment: SAED patterns should be discussed and compared in detail. The authors need to expand on the changes in the patterns at different irradiation exposures?
Response: The authors appreciate the comments provided by the reviewer and have made the according changes to the revised manuscript.
Page 6, lines 207-210
This is evidenced by the shielded region (Figure 2(b), region marked A) producing an SAED pattern with diffuse and indistinct rings, while the unshielded region (Figure 2(b), region marked B) exhibited a distinct diffraction pattern that could be attributed to cubic ZrO2.
Page 7, lines 237-240
Specifically, evidence of recrystallization was observed from TEM analysis after irradiating the film to a fluence of 3.13x1014 ions/cm2 and a d102 ring corresponding to tetragonal ZrO2 was observed after irradiating to a fluence of 3.13x1014 ions/cm2, 1.88x1015 ions/cm2, and 3.13x1015 ions/cm2 [41].
Page 9, lines 284-286
Rings corresponding to anatase (101) and a rutile (101) and (111) were indexed in the RT irradiated sample (Figure 5(f)), while only rings corresponding to rutile phase were visible in the SAED for the HT sample (Figure 5(i)).
Page 17, lines 578-580:
The SAED pattern in Figure 12a shows the initial amorphous diffraction pattern, and following electron irradiation, the SAED patterns in Figure 12d shows distinct rings that correspond with recrystallization.
- Comment: The article could be expanded to include more research articles for a more comprehensive review.
Response: Thanks for the comment. Though the authors would like to point out that the article is not meant to be a comprehensive review, but rather a perspective on work that has been published which demonstrates a-to-c transformation. This is why we believe that the current scope and covered materials matches well with our goals through this work.